# Role of Interleukins and New Perspectives in Mechanisms of Resistance to Chemotherapy in Gastric Cancer

**DOI:** 10.3390/biomedicines10071600

**Published:** 2022-07-05

**Authors:** Marlena Janiczek-Polewska, Łukasz Szylberg, Julian Malicki, Andrzej Marszałek

**Affiliations:** 1Department of Electroradiology, Poznan University of Medical Sciences, 61-701 Poznan, Poland; julian.malicki@wco.pl; 2Department of Clinical Oncology, Greater Poland Cancer Center, 61-866 Poznan, Poland; 3Department of Perinatology, Gynaecology and Gynaecologic Oncology, Collegium Medicum, Nicolaus Copernicus University, 85-067 Bydgoszcz, Poland; l.szylberg@cm.umk.pl; 4Department of Tumor Pathology and Pathomorphology, Oncology Centrer of Franciszek Łukaszczyk Memorial Hospital, 85-796 Bydgoszcz, Poland; 5Department of Oncologic Pathology, Prophylaxis Poznan University, Medical Sciences and Greater Poland Cancer Center, 61-866 Poznan, Poland; amars@ump.edu.pl

**Keywords:** gastric cancer, interleukins, cytokines, chemoresistance, tumor microenvironment

## Abstract

Gastric cancer (GC) is the fourth most common cancer in the world in terms of incidence and second in terms of mortality. Chemotherapy is the main treatment for GC. The greatest challenge and major cause of GC treatment failure is resistance to chemotherapy. As such, research is ongoing into molecular evaluation, investigating mechanisms, and screening therapeutic targets. Several mechanisms related to both the tumor cells and the tumor microenvironment (TME) are involved in resistance to chemotherapy. TME promotes the secretion of various inflammatory cytokines. Recent studies have revealed that inflammatory cytokines affect not only tumor growth, but also chemoresistance. Cytokines in TME can be detected in blood circulation and TME cells. Inflammatory cytokines could serve as potential biomarkers in the assessment of chemoresistance and influence the management of therapeutics in GC. This review presents recent data concerning research on inflammatory cytokines involved in the mechanisms of chemoresistance and provides new clues in GC treatment.

## 1. Introduction

Gastric cancer (GC) is the third leading cause of cancer death in men, the fifth in women, and fourth most common malignant tumor in the world [1]. About one fourth of the GC cases are diagnosed at the locally advanced gastric cancer (LAGC) stage. The 5-year survival rate is only from 20% to 26% in Europe, China, and the United States [2]. Alone, surgery treatment for LAGC did not produce satisfactory outcomes in these patients. Additional peri-operative chemotherapy improved the survival of patients with LAGC [3,4]. In 2006, the MAGIC study (a Phase III, randomized, controlled study) conducted a trial of patients with perioperative chemotherapy. The study group received three cycles of epirubicin, cisplatin, and fluorouracil (ECF) before, and after surgery. The control group received surgery alone. The study found that the surgery alone had a worse outcome than surgery with perioperative chemotherapy. Furthermore, the FLOT4 trial results (phase II/III multicenter, randomized controlled trial) showed that the stronger chemotherapy FLOT regimen was even better than the ECF/ECX regimen in terms of overall survival (OS) [5]. Palliative chemotherapy in patients with inoperable GC prolongs survival and improves its quality compared to symptomatic treatment [6]. In some patients with borderline unresectable tumors, the use of a systemic treatment may allow for resection with the assumption of radicality (another surgery after tumor remission is achieved—the stage is reduced) [7]. The choice of a treatment regimen should be based on the patient’s overall assessment, performance status, and side-effect profile. The most effective chemotherapy regimens include the combination of a platinum salt (cisplatin or oxaliplatin) with fluoropyrimidine and epirubicin [8]. Adding docetaxel to a regimen containing fluorouracil (FU) and cisplatin (DCF) prolongs survival, compared to double therapy, but is associated with greater toxicity [9]. If the overexpression or amplification of the HER2 receptor in GC cells is demonstrated in molecular tests, it is advisable to add trastuzumab to the regimen containing cisplatin and fluoropyrimidine in the 1-line treatment. Using trastuzumab in the GC study confirmed that chemotherapy combined with HER-2 targeted therapy resulted in a better therapeutic effect than chemotherapy alone for patients with high HER-2 expression [10]. 

Polymorphisms, gene mutations, and a unique genetic background can lead to different rates of response to the same chemotherapy regimen. However, the main reason for treatment failure is resistance to chemotherapy [11].

The mechanisms of drug resistance to chemotherapy are complex [12]. Several mechanisms related to both the tumor cells and the tumor microenvironment (TME) are involved in resistance to chemotherapy [13]. TME promotes the secretion of various inflammatory cytokines. Cytokines are a group of functional proteins that are promoted by the immune system. Cytokines play the role of key mediators of cellular communication in the TME. Moreover, they are important therapeutic targets and prognostic factors. Cancer cells can produce various cytokines and their receptors. Cytokines help cancer cells grow, survive and spread cancer [14,15]. Studies have shown that inflammatory cytokines affect chemoresistance [16].

Significant progress has been made in the study of the molecular mechanisms of drug resistance, but the technique used to detect and control drug resistance is still lacking in clinical practice. In this article, we would like to summarize and analyze the research progress in the field of the influence of interleukins in the chemoresistance of GC.

## 2. The Role of Interleukins in the Mechanism of Chemoresistance in Gastric Cancer

The TME plays a significant role in tumorigenesis and cancer progression. The cell types within the TME include bone marrow-derived cells, endothelial cells, cancer-associated fibroblasts (CAFs), and mesenchymal stem cells [12,13,15]. Immune cells may promote cancer initiation by secreting cytokines, which stimulate epithelial proliferation and generate DNA damage. Interleukins are a special group of cytokines. They are low-molecular-weight proteins. They are involved in the functioning of both the adaptive and innate immune systems. Interleukins regulate various steps of tumorigenesis. Moreover, recent reports indicate that interleukins in the TME can regulate cancer self-renewal and survival in various ways, sequentially promoting resistance to chemotherapy [14,15,16]. 

### 2.1. Interleukin 6

Interleukin-6 (IL-6) is a multi-faceted cytokine that mediates responses to trauma, infection, autoimmune diseases and is involved in the development of neoplastic diseases [17]. Cancer cells, as well as inflammatory and stromal cells, can produce IL-6. Its overexpression has been reported in many types of tumors [18]. The high level of IL-6 in the tumor microenvironment indicates a strong relationship between inflammation and cancer [19]. IL-6 in the tumor microenvironment promotes tumor formation by regulating all the hallmarks of cancer and many signaling pathways including metabolism, apoptosis, survival, proliferation, angiogenesis, invasiveness and metastasis. Additionally, IL-6 protects cancer cells from therapy-induced DNA damage, oxidative stress and apoptosis. This allows the repair and induction of antioxidant and anti-apoptotic pathways [20]. Accordingly, blocking IL-6 or inhibiting related signaling pathways, independently or in combination with conventional anti-cancer therapies, may be a potential therapeutic strategy in the treatment of cancer [21]. Several studies have also investigated the function of IL-6 in promoting resistance to chemotherapeutic agents in a variety of cancers, including GC [22,23]. IL-6 secreted by CAF, worked by activating the Jak1-STAT3 signaling pathway in the mechanism of chemoresistance in GC. The expression of IL-6 in biopsy specimens from patients treated with chemotherapy before surgery is significantly correlated with a poor response to chemotherapy in GC patients. Hence, treatment with an anti-IL-6 receptor monoclonal antibody (tocilizumab), in combination with chemotherapy, could serve as an appropriate strategy to improve chemotherapeutic efficacy by suppressing interactions between stromal cells and GC [23] (Table 1) (Figure 1).

### 2.2. Interleukin 8

Interleukin-8 (IL-8, CXCL8) was originally described as a chemokine. Its main function is to attract multinucleated inflammatory leukocytes [24]. Tumors very often produce this chemokine, which in this context performs various pro-tumor functions. These include angiogenesis, survival signaling for cancer stem cells, attraction of bone marrow cells endowed with immunosuppression, and local delivery of growth factors [25]. IL-8 is mainly produced by cancer cells; therefore, IL-8 serum levels have been shown to correlate with tumor advancement [26], and can be a useful biomarker for the detection of response to immunotherapy treatment [26,27]. In addition, IL-8 plays a significant role in tumor progression, which allowed for the development of several therapeutic strategies aimed at interfering with its functions and represents a promising therapeutic combination in the field of cancer immunotherapy [28]. Kuai et al. investigated the relationship between IL-8 and proliferation, adhesion, migration, invasion and chemosensitivity of GC cells. They used the human GC cell lines MKN-45 and KATO-III. IL-8 cDNA was stably transfected into human GC cell line MKN-45 and KATO-III. The expression of IL-8 in human GC cell line KATO-III was inhibited by RNA interference. Kuai et al. showed that the overexpression of IL-8 resulted in significant resistance to oxaliplatin in MKN-45 cells and the inhibition of IL-8 expression with a small amount of interfering RNA decreased oxaliplatin resistance in KATO-III cells. Moreover, the overexpression of IL-8 resulted in increased cell adhesion, migration and, invasion [29] (Table 1). In another study, Zhai et al. confirmed the significant role of IL-8 in the mechanism of platinum-based therapy resistance in GC. They showed that high levels of IL-8 in serum before therapy in patients with GC were associated with a poor response to platinum-based therapy. The level of IL-8 slowly increased during neoadjuvant chemotherapy and decreased after radical GC surgery. Immunohistochemistry has shown a strong expression of IL-8 in CAF in GC tissues in chemotherapy-resistant patients. Primary CAFs produced more IL-8 than the corresponding normal fibroblasts, and the human gastric fibroblast line Hs738 secreted more IL-8 when co-cultured with conditioned media from AGS or MGC-803 cells. IL-8 increased the IC50 of cisplatin (CDDP) in AGS or MGC-803 cells in vitro. IL-8 promotes chemoresistance in GC by activating PI3K, phosphorylated-AKT (*p*-AKT), phosphorylated-IKb (*p*-IKb), phosphorylated-p65 (*p*-p65), and upregulating ATP-binding cassette subfamily B member 1 (ABCB1). ABCB1 pumps cause the efflux of chemotherapeutic agents in cancers [30] (Table 1, Figure 1). Limpakan et al. also demonstrated that patients with low IL-8 levels were more sensitive to platinum drugs (i.e., cisplatin and oxaliplatin), compared to patients with high IL-8 levels. They obtained material from the biopsies performed during the endoscopic examinations before and/or after chemotherapy treatment, according to the FOLFOXIV regimen (oxaliplatin + 5-Fluorouracil (5-Fu) + leucovorin), and created primary gastric cultures. A 3-(4,5-dimethylthiazol-2-yl)-2,5-diphenyltetrazolium bromide (MTT) assay was performed to test ex vivo chemical susceptibility to cisplatin, oxaliplatin, 5-FU and irinotecan, and an enzyme-linked immunosorbent assay (ELISA) was performed to test cytokine levels. However, the authors of the study emphasized that the study was a pilot study with many limitations, including a small study group, and that the study focused solely on the ethnic population of northern Thailand [31] (Table 1). Moreover, the research was conducted on IL-8 using RNAseq analysis. RNA sequencing revealed significant molecular changes in drug resistance, in which IL-8 expression increased with increasing drug resistance. First-line chemotherapy (5-Fu and oxaliplatin) effectively inhibits GC growth but develops drug resistance. Second-line chemotherapy (paclitaxel monotherapy) reverses resistance to first-line chemotherapy for GC. To test the effect of targeting IL-8 on tumor-growth inhibition, reparixin was chosen. It is a small molecule inhibitor that manipulates IL-8/IL8R signaling to reduce IL-8 function. In vivo and in vitro experiments showed that IL8-targeted by RNA interference or reparixin reversed chemotherapy resistance with limited toxicity. Consequently, it was shown that the sequential treatment with first and second-line chemotherapy followed by reparixin inhibited the growth of GC and reduced the toxicity of the treatment and prolonged survival of mice [32] (Table 1). Some limitations exist in this study, including no exploration of the biology of the IL-8 system in myeloid leukocyte populations present in immunodeficient mice, and IL-8 absence from the rodent genome [32] (Table 1). The above data suggest that the assessment of IL-8 levels in GC patients may be useful as a predictive biomarker in monitoring drug resistance in these patients. Research was conducted using different methods, e.g., immunohistochemistry, ELISA or RNAseq analysis, and different materials were used for testing, e.g., biopsy and serum. Consequently, the evaluation of the level of IL-8 can be a cost-effective and easy method to predict chemoresistance in GC.

### 2.3. Interleukin 11

Interleukin 11 (IL-11) is involved in various stages of tumor development, including proliferation, angiogenesis, resistance to radio and chemotherapy, and inhibition of apoptosis [33]. IL-11 works by triggering the JAK-STAT3 pathway [34]. Elevated expression of IL-11 has been demonstrated in various epithelial and hematopoietic neoplasms [35]. IL-11 is secreted by cancer cells and TMA, i.e., cancer-associated fibroblasts and myeloid cells [36]. Fibroblasts facilitate chemotherapeutic drug-resistance development through the secretion of IL-11, and this latter via its receptor IL-11R activates the gp130/ JAK/STAT3 pathway, which is an anti-apoptosis signaling pathway in GC cells. Therapeutic strategies targeting the IL-11 signaling pathway induced by fibroblasts may be a promising clinical strategy for overcoming drug-resistant cancer [37] (Table 1, Figure 1).

### 2.4. Interleukin 24

Interleukin 24 (IL-24) is produced by immune cells such as bone marrow (inter alia proinflammatory macrophages) and lymphoid cells [38]. IL-24 affects immune cells, epithelial cells, and cancer cells. IL-24 is involved in immune response and anti-tumor activity [39,40]. IL-24 is involved in the apoptosis of neoplastic cells and the inhibition of angiogenesis [41]. Downstream effects of IL-24 upon binding to the IL-20 receptor can occur in a manner dependent or independent of the JAK/STAT pathway, which is involved in cytokine-mediated activities [42]. In tumor cells, apoptosis is induced independently of the JAK/STAT pathway after the exogenous addition of IL-24 [43]. In addition, IL-24 acts by binding to its molecular partners, including BiP/GRP78, Sig1R, IL-20 receptors, and dsRNA-dependent protein kinase (PKR) [44]. Mao et al. investigated the mechanism of chemosensitizing of IL-24 mediated by adenovirus (Ad-IL-24) gene therapy plus CDDP for MDR SGC7901/CDDP human GC cells in vitro and in vivo. The studies showed that the expression of IL-24 mRNA and protein was reduced in SGC7901/CDDP cells. Ad-IL-24 induced G2/M cell-cycle arrest, apoptosis and tumor suppression of SGC7901/CDDP cells in vitro and in vivo. In addition, Ad-IL-24 was related to up-regulation of Bax and down-regulation of *p*-gp and Bcl-2 in SGC7901/CDDP cells in vitro and in vivo. Consequently, these studies indicated that overexpression of the IL-24 gene can significantly promote chemosensitivity in GC cells with the MDR SGC7901/CDDP phenotype [45] (Table 1, Figure 1). These are the only studies on the role of IL-24 in the mechanisms of chemoresistance in GC. The results are promising, but more research is needed to conclusively assess the effect of IL-24 on chemoresistance.

### 2.5. Interleukin 33

Interleukin-33 (IL-33) is a member of the IL-1 gene family [46]. IL-33 is mainly ex-pressed in the nuclei of tissue lining cells, stromal cells, and activated marrow cells [47]. IL-33 is a damage-related molecular pattern (DAMP) molecule [48]. It plays a significant role in a variety of physiological and pathological situations. It takes part in allergies, inflammatory processes, autoimmune diseases, infectious diseases, and cancer [49]. The biological functions of IL-33 include maintaining tissue homeostasis, enhancing cellular immune responses, and mediating fibrosis during chronic inflammation. IL-33 affected tumor cells, promoting tumorigenesis, proliferation, survival, and metastasis. Moreover, it promotes tumor growth and metastasis by remodeling TME [50]. IL-33 acts on the ST2 receptor. The ST2 receptor is expressed by regulatory T cells (Treg), innate group 2 lymphoid cells (ILC2), bone marrow cells, cytotoxic NK cells, Th2 cells, Th1 and CD8 + T cells [51]. Ye et al. indicated that low concentrations of IL-33 protected against platinum-induced apoptosis in various GC cell lines, but not in normal gastric epithelial cells, and promotes cell invasion by activating the JNK pathway in GC cells. Moreover, IL-33 at concentrations below 100 pg/mL did not alter the rate of inhibition of GC cells [52] (Table 1, Figure 1).

**Figure 1 biomedicines-10-01600-f001:**
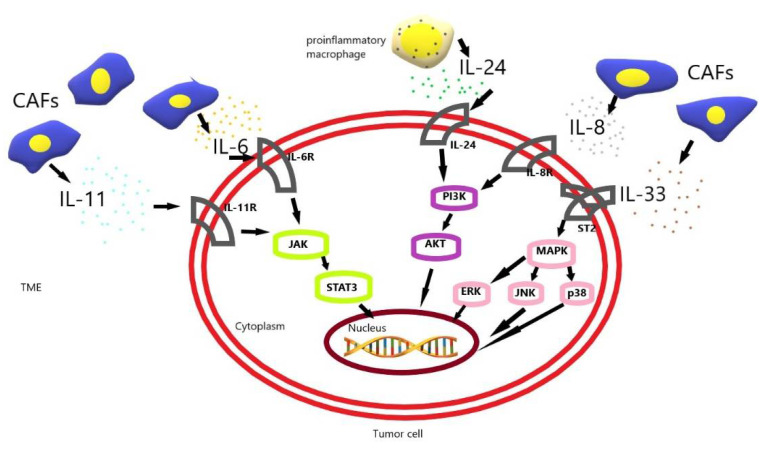
The schematic diagram of drug resistance development induced by IL-6, IL-8, IL-11, IL-24, IL-33 in gastric cancer cells [23,30,37,45,52]. CAF, cancer-associated fibroblast; TME, tumor microenvironment.

CAF-induced interleukin 6 (IL-6) and interleukin 11 (IL-11) activate the JAK-STAT3 pathway in gastric cancer (GC) cells. CAF-induced interleukin 33 (IL-33) activates the MAPK-ERK/JNK/p38 pathway in GC cells. Additionally, CAF-induced interleukin 8 (IL-8) and proinflammatory macrophage-induced interleukin 24 (IL-24) activate the PI3K-AKT pathway in GC cells. These pathways allow tumor cells to resist apoptosis and increase their survival, and their resistance to chemotherapy.

**Table 1 biomedicines-10-01600-t001:** Mechanisms of chemoresistance interleukins in treatment of GC.

Author	Year	Interleukin	Feature	Test Chemotherapy	Result	Reference
I.Ham et al.	2019	IL-6	increased production	5-fluorouracil Cisplatin	promotion of chemoresistance	[23]
W.Kuai et al.	2012	IL-8	increased production	Oxaliplatin	promotion of chemoresistance	[29]
J.Zhai et al.	2019	IL-8	increased production	Cisplatin	promotion of chemoresistance	[30]
S. Limpakan et al.	2019	IL-8	decreased production	5-fluorouracil oxaliplatin cisplatin irinotecan	increased chemosensitivity	[31]
H.Jiang et al.	2022	IL-8	Increased production	5-fluorouracil oxaliplatin paxlitaxel	reduction in chemotherapy resistance	[32]
J.Ma et al.	2019	IL-11	increased production	cisplatin doxorubicine etoposide	Increased chemoresistance	[37]
Z.Mao et al.	2013	IL-24	increased production	Cisplatin	promotion of chemosensitivity	[45]
X.L.Ye et al.	2015	IL-33	increased production	5-fluorouracil oxaliplatin cisplatin docetaxel	promotion of chemoresistance	[52]

## 3. The Role of Interleukins in the Development of Gastric Cancer—A New Perspective in the Mechanism of Chemoresistance

The risk of developing GC is increased by autoimmune and chronic gastritis caused by Helicobacter pylori infection [53]. Cytokines are secreted by immune cells and epithelial cells during chronic gastritis [54]. Some pro-inflammatory and anti-inflammatory cytokines contribute to tumor growth [55]. In patients with advanced GC, most cytokines that enhance or suppress the host anti-tumor immunity appear to have elevated levels of expression in the serum and the tumor environment [56]. Cancer progression and chemoresistance are closely related to the extracellular environment, which acquires characteristic properties during the development of cancer [15]. Cytokines stimulate the growth of cancer cells, accelerate chemoresistance, and promote tumor progression and recurrence [57]. Recent studies revealed that interleukin 1B, 2, 4, 10, 17A and 17F were involved in the development of GC and may potentially be involved in chemoresistance.

### 3.1. Interleukin 1 Beta

Interleukin 1 beta (IL-1B) is produced and secreted by immune cells, fibroblasts, or cancer cells [58]. IL-1B has a pleiotropic effect in cancer. It has a pro-tumor role but may also contribute to the anti-tumor immune response [59]. IL-1B signaling requires the type 1 IL-1 receptor antagonist (IL-1RA) [60]. This cytokine is an important mediator of the inflammatory response, cell proliferation, differentiation, and apoptosis in cancer. It is also involved in the mechanism of resistance to chemotherapy [61,62] (Table 2). The role of the IL-1B polymorphism in GC development is ambiguous, which is confirmed by the research presented below. Gonzales et al. did not show a relationship between the cytokine gene polymorphisms and the development of GC. They studied the influence of Helicobacter pylori infection and host genetic factors on GC risk. However, the research was conducted only on the Spanish population. DNA material from 404 GC patients and the same number of healthy controls was tested. Next, DNA material was typed for several functional polymorphisms: IL-1B, TNF-alpha, LTA, IL-12p40, IL-4, IL-1RN, IL-10, and TGFB1 genes. They used PCR, RFLP, and TaqMan assays [63] (Table 2). In another study, Song et al. demonstrated that IL-1B rs1143634 polymorphism might be associated with a decreased risk of GC. Moreover, it may be a protective factor against GC [64]. Interestingly, the meta-analysis showed even more divergent results. Xue et al. in a meta-analysis investigated the role of IL-1B cluster gene polymorphisms at positions −511, −31, and +3954, as well as a variable number of IL-1RN receptor tandem-repeat polymorphisms in the susceptibility to GC. For the meta-analysis 18 studies were eligible for IL1B-511, 21 studies for IL1B-31, 10 studies for *IL1B*+3954, and 20 studies for IL1RN variable number tandem repeat genetic polymorphisms, respectively. The studies showed that *IL-1B* -511 T allele and IL-1 RN *2 VNTR are significantly associated with an increased risk of developing GC among Caucasians, but not among Asians or Hispanics [65] (Table 2). As it results from the above studies, the role of IL-1B in the development of GC cannot be clearly indicated. The above data suggest that it is dependent on the polymorphism of *IL-1B* genes and on race. Consequently, it is unlikely to be a good biomarker for the assessment of chemoresistance in GC. But we need more detailed research to clearly rule this out.

### 3.2. Interleukin 2

Interleukin 2 (IL-2) has a pleiotropic effect on the immune system. It is produced by antigen simulated CD4+ T cells, CD8+ cells, natural killer (NK) cells, and activated dendritic cells (DC) [66]. IL-2 was one of the first FDA-approved cytokines for metastatic melanoma and renal cell carcinoma. However, this therapy has limitations, IL-2 has a short in vivo half-life, severe toxicity at therapeutic doses, and induction of immunosuppressive responses through regulatory T cell expansion [67]. Wu et al. tested association *IL-2* G-330T (rs2069762) and *IL-4* T-168C (rs2070874) with the risk of GC in a case-control study. *IL2* G-330T with GT/TT genotypes had a significantly reduced risk of gastric cardia cancer, compared with the *GG* genotype. Moreover, heterozygous -168TC and combined -168TC/CC genotypes were associated with a significantly decreased GC risk comparison with the *IL4* -168TT genotype. The study indicates that *IL2* G-330T and *IL4* T-168C promoter polymorphisms may contribute to the development of gastric cardia cancer in Chinese populations [68] (Table 2). In another study, Shin et al. compared the genotype of *IL-2* gene polymorphism with the risk of gastric ulcers (GU), GC, and duodenal ulcers (DU) in Korean patients. Polymorphism of the *IL-2*-330 gene was analyzed by polymerase chain reaction. The study showed that *IL-2* genetic polymorphism did not affect the pathogenesis of GU, GC, and DU in Korean patients [69] (Table 2). The role of IL-2 in the development of GC is ambiguous. Based on the studies presented above, it is not possible to assess the influence of IL-2 on the development of GC. These studies were limited by a homogeneous population. However, these data may suggest that IL-2 affects GC development in the Chinese population and is not related to GC development in the Korean population. Due to divergent information and population variability, IL-2 is also unlikely to be a good candidate for research looking for a biomarker in GC. 

### 3.3. Interleukin 4

IL-4 is a multifunctional cytokine. It is produced by mast cells, basophils, a subset of activated T cells, eosinophils, and neutrophils [70]. IL-4 plays a critical role in the regulation of immune responses, activation of mediators of cell growth, resistance to apoptosis, gene activation and, differentiation [71]. IL-4 takes part in the development of asthma, allergic inflammation, and multiple types of cancer [72]. Lai et al. conducted a study of 123 GC patients on genetic polymorphisms of the *MK*, *IL-4*, *p16*, *p21*, and *p53* genes, and assessed their association with GC carcinogenesis. Results showed that there was significant association of genetic polymorphisms between GC and control groups in *p53* genes, but no significant association with *IL-4* gene polymorphisms [73] (Table 2). This is confirmed by Gonzales et al., in their research [63] (Table 2). In another study, Omar et al. assessed polymorphisms of the *IL-1*, *IL-4*, *IL-6*, *and IL-10* gene cluster in a population of upper gastrointestinal cancers, including gastric cardia and non-cardia adenocarcinomas, esophageal squamous cell carcinoma and adenocarcinoma and in frequency-matched controls. *IL-10* was associated with the risk of non-cardia GC. They revealed that *IL-4* and *IL-6* were not associated with any of the tested tumors [74] (Table 2). In contrast, Yun et al. showed that the *IL-4* rs2243250 CC genotype and CT+CC genotype were associated with GC risk in the Chinese population, and *IL-4* haplotypes play an important role in the development of GC [75] (Table 2). Moreover, Comanzo et al. showed that *IL-10* (−819 C/T, rs1800871) promoters were associated with a lower risk for GC in a Mexican population, but they did not find significant association between *IL-4* –590 T/C (rs1800629), *IL-10* –592C/A (rs1800872), *IL-10* –1082A/G (rs1800896) with GC [76] (Table 2). Subsequent researchers focused on the role of IL-4 with the receptor and/or IL-13 in the development of GC [77,78,79]. He et al. did not observe in a case-control study about association between IL-4/IL-4R and genetic variations, GC risk, and their prognostic values [77] (Table 2); however, interestingly in another study, Noto et al. showed that IL4/IL13 signaling via IL4Ra regulate metaplasia in GC [78] (Table 3). This is also confirmed by Song et al. in their review [79]. Unfortunately, despite many studies, it is still not possible to clearly assess the role of IL-4 in the development of GC. Research suggests that IL4/IL13 signaling via IL4Ra may play a significant role in the development of GCs. Additionally, the effect on metaplasia may be the starting point for research on chemoresistance in GC.

### 3.4. Interleukin 10

Interleukin-10 (IL-10) is an anti-inflammatory cytokine. It plays a crucial role in preventing inflammatory and autoimmune pathologies. IL-10 is secreted by tumor-associated macrophages TAMs and Treg cells in GC [80]. IL-10 promotes tumor proliferation by inhibiting immune responses in various malignancies. IL-10 acts by signaling pathways STAT3 and NF-κB [81]. CpG hypermethylation was associated with decreased IL-10 gene expression and that hypomethylation was associated with an increased gene in GC and adjacent tissue. Moreover, hypomethylation of CpG islands within the IL-10 gene in GC and adjacent tissue was associated with decreased OS and a worse prognosis [82] (Table 3). Furthermore, IL-10 was expressed in the cell-culture supernatant of GC TAMs, and that exposing tumor cells to this particular supernatant increased tumor proliferation. Additionally, anti-IL10 antibody partially blocked proliferation induced by the supernatant, supporting a link between IL-10 and tumor proliferation [83] (Table 3). The above data indicate an association of IL-10 in GC development. IL-10 influences the progression of neoplastic cells. Thus, it will be reasonable to suppose that IL-10 may play a role in chemoresistance in GC.

### 3.5. Interleukin 17A and Interleukin 17F

The Interleukin-17 (IL-17) family of cytokines consists of six structurally similar cytokines from IL-17A to F. IL-17F has similar homology to IL-17A [84]. They are expressed by T helper (Th)17 cells, CD8 T cells, natural killer (NK) cells, lymphoid tissue inducer (LTi) cells, type 3 innate lymphoid cells, and γδ T cells [85]. IL-17A and IL-17F bind to the same complex of IL-17RA and IL-17RC [86]. They take part in the development of various diseases such as autoimmune diseases, inflammatory diseases, and malignant neoplasms [87]. Liu et al. performed a meta-analysis about the association of *IL-17A rs2275913* and *IL-17F rs763780* polymorphisms with GC risk. A total of 3345 cases for *IL-17A* rs2275913 and 1784 cases for *IL-17F* rs763780 were included in the analysis. A meta-analysis showed that *IL-17A* rs2275913 A/G polymorphism and *IL-17F* rs763780 C/T polymorphism might be associated with an increased risk of GC in Asians [88] (Table 2). In another study, Wang et al. showed that IL-17A could promote the invasion of GC cells by activating the NF-κB pathway, and subsequently upregulating the expression of MMP-2 and MMP-9 [89] (Table 3). There are no data in the available literature on the role of IL-17A and IL-17F in GC chemoresistance. However, the above data suggest that these cytokines may play a significant role in the chemoresistance mechanism in GC.

**Table 2 biomedicines-10-01600-t002:** The role of polymorphism in interleukins in the development of GC.

Author	Year	Interleukin	SNPs	Result	Reference
Gonzales et al.	2007	IL-1B	polymorphisms in IL-1B	No significant association	[63]
Xue et al.	2010	IL-1B	*IL1B*-511 *IL1RN*	Increased GC risk	[65]
Xue et al.	2010	IL-1B	*IL1B*-31 *IL1B*+3954	No significant association	[65]
Song et al.	2021	IL-1B	*IL-1B* rs1143634	Decreased GC risk	[64]
Shin et al.	2008	IL-2	*IL-2*-330	No significant association	[69]
Wu et al.	2009	IL-2	*IL-2* G-330T rs2069762	Increased GC risk	[68]
Omar et al.	2003	IL-4	*IL-4* 590 C/T rs2243250	No significant association	[74]
Lai et al.	2005	IL-4	*IL-4* 590 rs2243250	No significant association	[73]
Gonzales et al.	2007	IL-4	*IL-4* 590 C>T rs2243250	No significant association	[63]
Wu et al.	2009	IL-4	*IL-4*T-168C rs2070874	Increased GC risk	[68]
Yun et al.	2017	IL-4	*IL-4* rs2243250	Increased GC risk	[75]
Martinez- Campos et al.	2019	IL-4	*IL-4*-590C/T rs2243250	No significant association	[76]
He et al.	2019	IL-4	*IL-4* rs2243248 *IL-4* rs2070874 *IL-4R* rs2057768 *IL-4R* rs2107356 *IL-4R* rs1805015 *IL-4R* rs1801275	No significant association	[77]
Martinez- Campos et al.	2019	IL-10	*IL-10*–819C/T rs1800871	Significant association	[76]
Martinez- Campos et al.	2019	IL-10	*IL-10*–592C/A rs1800872 *IL-10*–1082A/G rs1800896	No significant association	[76]
Liu et al.	2015	IL-17A	*IL-17A* rs2275913	Increased GC risk	[88]
Liu et al.	2015	IL-17F	*IL-17F* rs763780	Increased GC risk	[88]

SNPs: single nucleotide polymorphisms.

**Table 3 biomedicines-10-01600-t003:** The role of interleukins in the development of GC.

Author	Year	Interleukin	Result	Reference
Noto et al.	2022	IL-4	No significant association	[78]
Chen et al.	2019	IL-10	Increased GC proliferation	[83]
Tang et al.	2021	IL-10	Poor prognosis Decreased OS	[82]
Note et al.	2022	IL-13	Promotes metaplasia	[78]
Wang et al.	2014	IL-17A	Promotes invasion	[89]

## 4. Conclusions

Resistance to chemotherapy remains the greatest impediment in the treatment of cancer patients. Our understanding of the molecular mechanisms of chemoresistance is still incomplete. There is a wide range of molecular mechanisms of chemoresistance in GC. An important role in chemoresistance in GC is played by the dysregulation of cell-signaling pathways and the interactions between cancer cells and the tumor microenvironment. In the tumor microenvironment, cytokines and growth factors exhibit key functions in chemotherapeutic resistance by maintaining the activation of various survival-related signaling pathways. In addition, each of the studies had its limitations, which was also pointed out by some authors. The studies have contributed foundational insights, identifying several keys of interleukins that cancer cells commonly hyperactivate or block to evade the action of the chemotherapy. Most studies concerned IL-8. However, all interleukins in the studies have shown an effect on chemoresistance in the treatment of GC. Additionally, we need the development of diagnostic tools that are simple and affordable, which will enable the appropriate qualification of patients for chemotherapy. These conditions are met by GC biopsy material and cytokines present in it. Biopsy material is available from every patient with GC, as it is an essential element in making a diagnosis. As is evident from the above data, cytokines identified in the biopsy material may also prove applicable as chemoresistance biomarkers in GC. This approach of chemoresistance detection is not only minimally invasive and cost-effective but also could be easily accessible. These findings will be very helpful for the development of personalized therapies based on the prediction of the chemosensitivity of cancer cells in the future. Despite the promising results, further studies are needed to confirm the role of interleukins in the mechanisms of resistance to chemotherapy in GC.

## Data Availability

Not applicable.

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
