# Peer review of "Role of Interleukins and New Perspectives in Mechanisms of Resistance to Chemotherapy in Gastric Cancer"

_biomedicines, 2022, doi:10.3390/biomedicines10071600_

Round 1

Reviewer 1 Report

The manuscript entitled: "Role of interleukins in mechanisms of resistance to chemotherapy in gastric cancer", has focused on the role of various interleukins in gastric cancer chemoresistance. 

However, the manuscript is mainly "narrative", having used very few pertinent references per each interleukin and has made no conclusive and informative statement of their own.

Basically each paragraph describes a single study, with no interpretation or expert opinion of the  authors included.

In addition, none of the references belong to any of the authors to indicate their expertise and track record in this area.

Author Response

Thank you very much for your valuable comments. We have corrected the article as suggested.

  1. We improved the form of the article and added comments.
  2. Single studies are described in the review, as unfortunately there are currently few studies available in the literature on chemoresistance in gastric cancer. We wanted to focus only on this topic and this organ.
  3. Therefore, we have not included a link to our research on the role of interleukins in carcinogenesis. We investigated the role of interleukins in the development of colon cancer and prostate cancer.

List of articles:

  1. Alternative inflammatory cytokines pathways in prostate cancer: In search of new therapeutic options, Marlena Janiczek, Łukasz Szylberg, Paulina Antosik, Anna Kasperska, Andrzej Marszałek, Abstract, ASCO 2020, DOI:10.1200/JCO.2020.38.15_suppl.e17504 https://meetinglibrary.asco.org/record/187854/abstract
  2. Immunotherapy as a Promising Treatment for Prostate Cancer: A Systematic Review, Marlena Janiczek,Łukasz Szylberg, Anna Kasperska, Adam Kowalewski,Martyna Parol, Paulina Antosik, Barbara Radecka, Andrzej Marszałek, Journal of Immunology Research, Volume 2017 (2017), Article ID 4861570, 6 pages.
  3. Expression of COX-2, IL-1β, TNF-α and IL-4 in epithelium of serrated adenoma, adenoma and hyperplastic polyp, Łukasz Szylberg, Marlena Janiczek, Aneta Popiel, Andrzej Marszałek Arch Med Sci 2016 Feb 2;12(1):172-8.
  4. Large Bowel Genetic Background and Inflammatory Processes in Carcinogenesis--Systematic Review, Łukasz Szylberg, Marlena Janiczek, Aneta Popiel, Andrzej Marszałek, Adv Clin Exp Med 2015 Jul-Aug;24(4):555-63.
  5. Serrated Polyps and their Alternative Pathway to the Colorectal Cancer: A Systematic Review, Łukasz Szylberg, Marlena Janiczek, Aneta Popiel, and Andrzej Marszałek, Volume 2015, Gastroenterology Research and Practice.
  6. Impact of COX-2, IL-1β, TNF-α, IL-4 and IL-10 on the process of carcinogenesisin the large bowel, Andrzej Marszałek, Łukasz Szylberg, Ewa Wiśniewska, Marlena Janiczek, 04/2012, Polish Journal of Pathology.

Best regards,

Marlena Janiczek-Polewska

Reviewer 2 Report

I think is an interesting review for the field as interleukins play an important role in cancer.

The paper is really well organised. I would suggest:

-The title might be extended not only to resistance to chemotherapies but also to the development of the gastric cancer as in the second part the studies are more focus in the role of the development.

-The paragraph between lines 124 and 134 is a bit difficult to follow: at first it talks about one cell line, then it talks about other one, and then it comes back to the first one. I think there is not a proper flow in the narration.

-I think that in the conclussion part there is a lack on the focus of the use of interleukins as a possible biomarker in biopsies as it is a less invasive way of diagnosis in the patient

Author Response

Thank you very much for your valuable comments. We have corrected the article as suggested.

Best regards,

Marlena Janiczek-Polewska

Reviewer 3 Report

I have read your manuscript “Role of interleukins in mechanisms of resistance to chemotherapy in gastric cancer” with great pleasure. This is a really interesting paper, on a hot topic in pathology. The contribution is timely. However, the authors have to address the following before it can be considered.

Major:

1.     Please organize the third part " The role of interleukins in the development of gastric cancer - a new perspective in the mechanism of chemoresistance " into a table.

2.     After a duplication-check on your manuscript of biomedicines-1784939, there a few paragraphs/sentences are almost the same with the published papers (e.g., lines 30-45, 102-108, 137-146, 171-175, 179-184, 190-193, 194-206, 210-215, 223-227, 230-232, 242-247, 251-257, 293-297, 316-321, 353-363, 373-376), could you please rewrite/make revisions accordingly?

Minor:

1.     The citation style in the manuscript should be rechecked. For example, it should be [14, 15] instead of [14][15].

2.     Please change (Table 1) (Figure 1) to (Table 1, Figure 1).

3.     Please change interleukin-6 (IL-6) to IL-6 (line 103).

4.     In line 183 “by tha lack of”, please recheck.

Author Response

(The authors gave the same response as above.)

Round 2

Reviewer 1 Report

My previous comments still stand.

Author Response

We are sorry, but we are unable to make changes to the article if we do not know what is wrong with the first correction. We believe we have made all the expected changes to the first revision. Kind regards, Marlena Janiczek-Polewska

Reviewer 3 Report

The manuscript can be accepted in present form.

Author Response

Thank you very much. Kind regards, Marlena Janiczek-Polewska